# NETWORK OF GRAPH CONVOLUTIONAL NETWORKS TRAINED ON RANDOM WALKS

## ABSTRACT

Graph Convolutional Networks (GCNs) are a recently proposed architecture which has had success in semi-supervised learning on graph-structured data. At the same time, unsupervised learning of graph embeddings has benefited from the information contained in random walks. In this paper we propose a model, Network of GCNs (N-GCN), which marries these two lines of work. At its core, N-GCN trains multiple instances of GCNs over node pairs discovered at different distances in random walks, and learns a combination of the instance outputs which optimizes the classification objective. Our experiments show that our proposed N-GCN model achieves state-of-the-art performance on all of the challenging node classification tasks we consider: Cora, Citeseer, Pubmed, and PPI. In addition, our proposed method has other desirable properties, including generalization to recently proposed semi-supervised learning methods such as GraphSAGE, allowing us to propose N-SAGE, and resilience to adversarial input perturbations.

## 1 INTRODUCTION

Semi-supervised learning on graphs is important in many real-world applications, where the goal is to recover labels for all nodes given only a fraction of labeled ones. Some applications include social networks, where one wishes to predict user interests, or in health care, where one wishes to predict whether a patient should be screened for cancer. In many such cases, collecting node labels can be prohibitive. However, edges between nodes can be easier to obtain, either using an explicit graph (e.g. social network) or implicitly by calculating pairwise similarities (e.g. using a patient-patient similarity kernel, Merdan et al., 2017).

Convolutional Neural Networks (LeCun et al., 1998) learn location-invariant hierarchical filters, enabling significant improvements on Computer Vision tasks (Krizhevsky et al., 2012; Szegedy et al., 2015; He et al., 2016). This success has motivated researchers (Bruna et al., 2014) to extend convolutions from spatial (i.e. regular lattice) domains to graph-structured (i.e. irregular) domains, yielding a class of algorithms known as Graph Convolutional Networks (GCNs).

Formally, we are interested in semi-supervised learning where we are given a graph $\mathcal{G} = (\mathcal{V}, \mathcal{E})$ with $N = |\mathcal{V}|$ nodes; adjacency matrix $A$; and matrix $X \in \mathbb{R}^{N \times F}$ of node features. Labels for only a subset of nodes $\mathcal{V}_L \subset \mathcal{V}$ observed. In general, $|\mathcal{V}_L| \ll |\mathcal{V}|$. Our goal is to recover labels for all unlabeled nodes $\mathcal{V}_U = \mathcal{V} - \mathcal{V}_L$, using the feature matrix $X$, the known labels for nodes in $\mathcal{V}_L$, and the graph $G$. In this setting, one treats the graph as the "unsupervised" and labels of $\mathcal{V}_L$ as the "supervised" portions of the data.

Depicted in Figure 1, our model for semi-supervised node classification builds on the GCN module proposed by Kipf & Welling (2017), which operates on the normalized adjacency matrix $\hat{A}$, as in GCN$(\hat{A})$, where $\hat{A} = D^{-\frac{1}{2}} A D^{-\frac{1}{2}}$, and $D$ is diagonal matrix of node degrees. Our proposed extension of GCNs is inspired by the recent advancements in random walk based graph embeddings (e.g. Perozzi et al., 2014; Grover & Leskovec, 2016; Abu-El-Haija et al., 2017). We make a Network of GCN modules (N-GCN), feeding each module a different power of $\hat{A}$, as in $\{\text{GCN}(\hat{A}^0), \text{GCN}(\hat{A}^1), \text{GCN}(\hat{A}^2), \dots\}$. The $k$-th power contains statistics from the $k$-th step of a random walk on the graph. Therefore, our N-GCN model is able to combine information from various step-sizes. We then combine the output of all GCN modules into a classification sub-network, and we jointly train all GCN modules and the classification sub-network on the upstream objective,

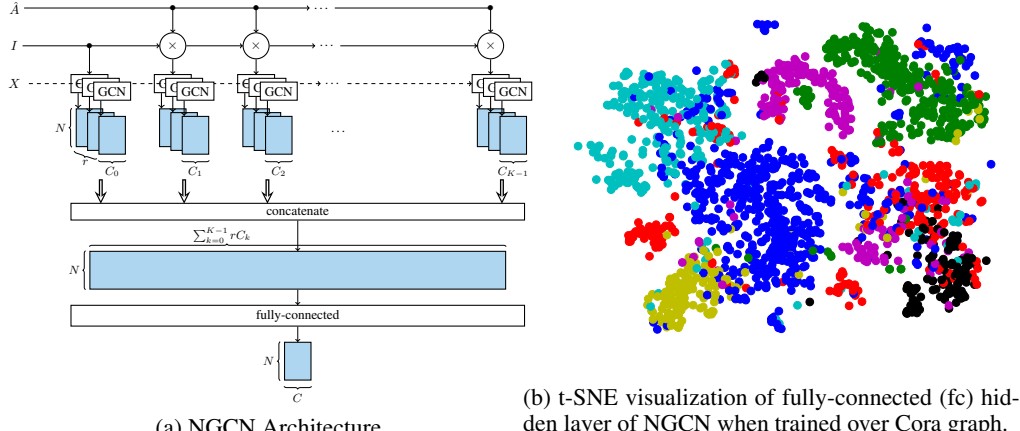

(a) NGCN Architecture

(b) t-SNE visualization of fully-connected (fc) hidden layer of NGCN when trained over Cora graph.

Figure 1: Left: Model architecture, where $\hat{A}$ is the normalized normalized adjacency matrix, $I$ is the identity matrix, $X$ is node features matrix, and $\times$ is matrix-matrix multiply operator. We calculate $K$ powers of the $\hat{A}$, feeding each power into $r$ GCNs, along with $X$. The output of all $K \times r$ GCNs can be concatenated along the column dimension, then fed into fully-connected layers, outputting $C$ channels per node, where $C$ is size of label space. We calculate cross entropy error, between rows *prediction* $N \times C$ with known labels, and use them to update parameters of classification sub-network and all GCNs. Right: pre-relu activations after the first fully-connected layer of a 2-layer classification sub-network. Activations are PCA-ed to 50 dimensions then visualized using t-SNE.

semi-supervised node classification. Weights of the classification sub-network give us insight on how the N-GCN model works. For instance, in the presence of input perturbations, we observe that the classification sub-network weights shift towards GCN modules utilizing higher powers of the adjacency matrix, effectively widening the "receptive field" of the (spectral) convolutional filters. We achieve state-of-the-art on several semi-supervised graph learning tasks, showing that explicit random walks enhance the representational power of vanilla GCN's.

The rest of this paper is organized as follows. Section 2 reviews background work that provides the foundation for this paper. In Section 3, we describe our proposed method, followed by experimental evaluation in Section 4. We compare our work with recent closely-related methods in Section 5. Finally, we conclude with our contributions and future work in Section 6.

## 2 BACKGROUND

### 2.1 SEMI-SUPERVISED NODE CLASSIFICATION

Traditional label propagation algorithms (Weston et al., 2012; Belkin et al., 2006a) learn a model that transforms node features into node labels and uses the graph to add a regularizer term:

$$\mathcal{L}_{\text{label.propagation}} = \mathcal{L}_{\text{classification}} + \mathcal{L}_{reg} = \mathcal{L}_{\text{classification}} + \lambda f(X)^T \Delta f(X), \tag{1}$$

where $f : \mathbb{R}^{N \times d_0} \to \mathbb{R}^{N \times C}$ is the model, $\Delta$ is the graph Laplacian, and $\lambda \in \mathbb{R}$ is the regularization coefficient hyperparameter.

### 2.2 GRAPH CONVOLUTIONAL NETWORKS

Graph Convolution (Bruna et al., 2014) generalizes convolution from Euclidean domains to graph-structured data. Convolving a "filter" over a signal on graph nodes can be calculated by transforming both the filter and the signal to the Fourier domain, multiplying them, and then transforming the result back into the discrete domain. The signal transform is achieved by multiplying with the eigenvectors of the graph Laplacian. The transformation requires a quadratic eigendecomposition of the symmetric Laplacian; however, the low-rank approximation of the eigendecomposition can be calculated using truncated Chebyshev polynomials (Hammond et al., 2011). For instance, Kipf &

Welling (2017) calculates a rank-1 approximation of the decomposition. They propose a multi-layer Graph Convolutional Networks (GCNs) for semi-supervised graph learning. Every layer computes the transformation:

$$H^{(l+1)} = \sigma\left(\hat{A}H^{(l)}W^{(l)}\right),\tag{2}$$

where $H^{(l)} \in \mathbb{R}^{N \times d_l}$ is the input activation matrix to the $l$-th hidden layer with row $H_i^{(l)}$ containing a $d_l$-dimensional feature vector for vertex $i \in \mathcal{V}$, and $W^{(l)} \in \mathbb{R}^{d_l \times d_{l+1}}$ is the layer's trainable weights. The first hidden layer $H^{(0)}$ is set to the input features $X$. A softmax on the last layer is used to classify labels. All layers use the same "normalized adjacency" $\hat{A}$, obtained by the "renormalization trick" utilized by Kipf & Welling (2017), as $\hat{A} = D^{-\frac{1}{2}}AD^{-\frac{1}{2}}$.[1]

Eq. (2) is a first order approximation of convolving filter $W^{(l)}$ over signal $H^{(l)}$ (Hammond et al., 2011; Kipf & Welling, 2017). The left-multiplication with $\hat{A}$ averages node features with their direct neighbors; this signal is then passed through a non-linearity function $\sigma(\cdot)$ (e.g, ReLU$(z) = \max(0, z)$). Successive layers effectively *diffuse* signals from nodes to neighbors.

Two-layer GCN model can be defined in terms of vertex features $X$ and normalized adjacency $\hat{A}$ as:

$$\text{GCN}_{\text{2-layer}}(\hat{A}, X; \theta) = \text{softmax}\left(\hat{A}\sigma(\hat{A}XW^{(0)})W^{(1)}\right),\tag{3}$$

where the GCN parameters $\theta = \left\{W^{(0)}, W^{(1)}\right\}$ are trained to minimize the cross-entropy error over labeled examples. The output of the GCN model is a matrix $\mathbb{R}^{N \times C}$, where $N$ is the number of nodes and $C$ is the number of labels. Each row contains the label scores for one node, assuming there are $C$ classes.

## 2.3 GRAPH EMBEDDINGS

Node Embedding methods represent graph nodes in a continuous vector space. They learn a dictionary $Z \in \mathbb{R}^{N \times d}$, with one $d$-dimensional embedding per node. Traditional methods use the adjacency matrix to learn embeddings. For example, Eigenmaps (Belkin & Niyogi, 2003) calculates the following constrained optimization:

$$\sum_{i,j} ||A_{ij}(Z_i - Z_J)|| \text{ s.t. } Z^TDZ = I,\tag{4}$$

where $I$ is identity vector. Skipgram models on text corpora (Mikolov et al., 2013) inspired modern graph embedding methods, which simulate random walks to learn node embeddings (Perozzi et al., 2014; Grover & Leskovec, 2016). Each random walk generates a sequence of nodes. Sequences are converted to textual paragraphs, and are passed to a word2vec-style embedding learning algorithm (Mikolov et al., 2013). As shown in Abu-El-Haija et al. (2017), this learning-by-simulation is equivalent, in expectation, to the decomposition of a random walk co-occurrence statistics matrix $\mathcal{D}$. The expectation on $\mathcal{D}$ can be written as:

$$\mathbb{E}[\mathcal{D}] \propto \mathbb{E}_{q \sim \mathcal{Q}}\left[(\mathcal{T})^q\right] = \mathbb{E}_{q \sim \mathcal{Q}}\left[\left(D^{-1}A\right)^q\right],\tag{5}$$

where $\mathcal{T} = D^{-1}A$ is the row-normalized transition matrix (a.k.a right-stochastic adjacency matrix), and $\mathcal{Q}$ is a "context distribution" that is determined by random walk hyperparameters, such as the length of the random walk. The expectation therefore weights the importance of one node on another as a function of how well-connected they are, and the distance between them. The main difference between traditional node embedding methods and random walk methods is the optimization criteria: the former minimizes a loss on representing the adjacency matrix $A$ (see Eq. 4), while the latter minimizes a loss on representing random walk co-occurrence statistics $\mathcal{D}$.

## 3 OUR METHOD

### 3.1 MOTIVATION

Graph Convolutional Networks and random walk graph embeddings are individually powerful. Kipf & Welling (2017) uses GCNs for semi-supervised node classification. Instead of following tradi-

---

[1]with added self-connections added as $A_{ii} = 1$, similar to Kipf & Welling (2017)

tional methods that use the graph for regularization (e.g. Eq. 4), Kipf & Welling (2017) use the adjacency matrix for training and inference, effectively diffusing information across edges at all GCN layers (see Eq. 6). Separately, recent work has showed that random walk statistics can be very powerful for learning an unsupervised representation of nodes that can preserve the structure of the graph (Perozzi et al., 2014; Grover & Leskovec, 2016; Abu-El-Haija et al., 2017).

Under special conditions, it is possible for the GCN model to learn random walks. In particular, consider a two-layer GCN defined in Eq. 6 with the assumption that first-layer activation is identity as $\sigma(z) = z$, and weight $W^{(0)}$ is an identity matrix (either explicitly set or learned to satisfy the upstream objective). Under these two identity conditions, the model reduces to:

$$\text{GCN}_{\text{2-layer-special}}(\hat{A}, X) = \text{softmax}\left(\hat{A}\hat{A}XW^{(1)}\right) = \text{softmax}\left(\hat{A}^2 XW^{(1)}\right), \tag{6}$$

where $\hat{A}^2$ can be expanded as:

$$\hat{A}^2 = \left(D^{-\frac{1}{2}}AD^{-\frac{1}{2}}\right)\left(D^{-\frac{1}{2}}AD^{-\frac{1}{2}}\right) = D^{-\frac{1}{2}}A\left[D^{-1}A\right]D^{-\frac{1}{2}} = D^{-\frac{1}{2}}A\mathcal{T}D^{-\frac{1}{2}}. \tag{7}$$

By multiplying the adjacency $A$ with the transition matrix $\mathcal{T}$ before normalization, the GCN is effectively doing a one-step random walk.

## 3.2 EXPLICIT RANDOM WALKS

The special conditions described above are not true in practice. Although stacking hidden GCN layers allows information to flow through graph edges, this flow is *indirect* as the information goes through feature reduction (matrix multiplication) and a non-linearity (activation function $\sigma(\cdot)$). Therefore, the vanilla GCN cannot directly learn high powers of $\hat{A}$, and could struggle with modeling information across distant nodes. We hypothesize that making the GCN directly operate on random walk statistics will allow the network to better utilize information across distant nodes, in the same way that node embedding methods (e.g. DeepWalk, Perozzi et al. (2014)) operating on $\mathcal{D}$ are superior to traditional embedding methods operating on the adjacency matrix (e.g. Eigenmaps, Belkin & Niyogi (2003)). Therefore, in addition to feeding only $\hat{A}$ to the GCN model as proposed by Kipf & Welling (2017) (see Eq. 6), we propose to feed a $K$-degree polynomial of $\hat{A}$ to $K$ instantiations of GCN. Generalizing Eq. (7) gives:

$$\hat{A}^k = D^{-\frac{1}{2}}A\mathcal{T}^{k-1}D^{-\frac{1}{2}}. \tag{8}$$

We also define $\hat{A}^0$ to be the identity matrix. Similar to Kipf & Welling (2017), we add self-connections and convert directed graphs to undirected ones, making $\hat{A}$ and hence $\hat{A}^k$ symmetric matrices. The eigendecomposition of symmetric matrices is real. Therefore, the low-rank approximation of the eigendecomposition Hammond et al. (2011) is still valid, and a one layer of Kipf & Welling (2017) utilizing $\hat{A}^k$ should still approximate multiplication in the Fourier domain.

## 3.3 NETWORK OF GCNS

Consider $K$ instantiations of $\{\text{GCN}(\hat{A}^0, X), \text{GCN}(\hat{A}^1, X), \ldots, \text{GCN}(\hat{A}^{K-1}, X)\}$. Each GCN outputs a matrix $\mathbb{R}^{N \times C_k}$, where the $v$-th row describes a latent representation of that particular GCN for node $v \in \mathcal{V}$, and where $C_k$ is the latent dimensionality. Though $C_k$ can be different for each GCN, we set all $C_k$ to be the same for simplicity. We then combine the output of all $K$ GCN and feed them into a classification sub-network, allowing us to jointly train all GCNs and the classification sub-network via backpropagation. This should allow the classification sub-network to choose features from the various GCNs, effectively allowing the overall model to learn a combination of features using the raw (normalized) adjacency, different steps of random walks, and the input features $X$ (as they are multiplied by identity $\hat{A}^0$).

### 3.3.1 FULLY-CONNECTED CLASSIFICATION NETWORK

From a deep learning prospective, it is intuitive to represent the classification network as a fully-connected layer. We can concatenate the output of the $K$ GCNs along the column dimension, i.e. concatenating all $\text{GCN}(X, \hat{A}^k)$, each $\in \mathbb{R}^{N \times C_k}$ into matrix $\in \mathbb{R}^{N \times C_K}$ where $C_K = \sum_k C_k$.

We add a fully-connected layer $f_{\text{fc}} : \mathbb{R}^{N \times C_K} \to \mathbb{R}^{N \times C}$, with trainable parameter matrix $W_{\text{fc}} \in \mathbb{R}^{C_K \times C}$, written as:

$$\text{N-GCN}_{\text{fc}}(\hat{A}, A; W_{\text{fc}}, \theta) = \text{softmax}\left(\left[\ \text{GCN}(\hat{A}^0, X; \theta^{(0)}) \ \vdots \ \text{GCN}(\hat{A}^1, X; \theta^{(1)}) \ \vdots \ \dots \ \right] W_{\text{fc}}\right). \quad (9)$$

The classifier parameters $W_{\text{fc}}$ are jointly trained with GCN parameters $\theta = \{\theta^{(0)}, \theta^{(1)}, \dots\}$. We use subscript fc on N-GCN to indicate the classification network is a fully-connected layer.

### 3.3.2 ATTENTION CLASSIFICATION NETWORK

We also propose a classification network based on "softmax attention", which learns a convex combination of the GCN instantiations. Our attention model (N-GCN$_\text{a}$) is parametrized by vector $\widetilde{m} \in \mathbb{R}^K$, one scalar for each GCN. It can be written as:

$$\text{N-GCN}_\text{a}(\hat{A}, X; m, \theta) = \sum_k m_k \text{GCN}(\hat{A}^k, X; \theta^{(k)}) \quad (10)$$

where $m$ is output of a softmax: $m = \text{softmax}(\widetilde{m})$.

This softmax attention is similar to "Mixture of Experts" model, especially if we set the number of output channels for all GCNs equal to the number of classes, as in $C_0 = C_1 = \cdots = C$. This allows us to add cross entropy loss terms on all GCN outputs in addition to the loss applied at the output NGCN, forcing all GCN's to be independently useful. It is possible to set the $m \in \mathbb{R}^K$ parameter vector "by hand" using the validation split, especially for reasonable $K$ such as $K \leq 6$. One possible choice might be setting $m_0$ to some small value and remaining $m_1, \dots, m_{K-1}$ to the harmonic series $\frac{1}{k}$; another choice may be linear decay $\frac{K-k}{K-1}$. These are respectively similar to the context distributions of GloVe (Pennington et al., 2014) and word2vec (Mikolov et al., 2013; Levy et al., 2015). We note that if on average a node's information is captured by its direct or nearby neighbors, then the output of GCNs consuming lower powers of $\hat{A}$ should be weighted highly.

### 3.4 TRAINING

We minimize the cross entropy between our model output and the known training labels $Y$ as:

$$\min \text{diag}(\mathcal{V}_L)\left[Y \circ \log \text{N-GCN}(X, \hat{A})\right], \quad (11)$$

where $\circ$ is Hadamard product, and $\text{diag}(\mathcal{V}_L)$ denotes a diagonal matrix, with entry at $(i, i)$ set to 1 if $i \in \mathcal{V}_L$ and 0 otherwise. In addition, we can apply intermediate supervision for the NGCN$_\text{a}$ to attempt make all GCN become independently useful, yielding minimization objective:

$$\min_{m, \theta} \text{diag}(\mathcal{V}_L)\left[Y \circ \log \text{N-GCN}_\text{a}(\hat{A}, X; m, \theta) + \sum_k Y \circ \log \text{GCN}(\hat{A}^k, X; \theta^{(k)})\right]. \quad (12)$$

### 3.5 GCN REPLICATION

To simplify notation, our N-GCN derivations (e.g. Eq. 9) assume that there is one GCN per $\hat{A}$ power. However, our implementation feeds every $\hat{A}$ to $r$ GCN modules, as shown in Fig. 1.

### 3.6 GENERALIZATION TO OTHER GRAPH MODELS

In addition to vanilla GCNs (e.g. Kipf & Welling, 2017), our derivation also applies to other graph models including GraphSAGE (SAGE, Hamilton et al., 2017). Algorithm 1 shows a generalization that allows us to make a network of arbitrary graph models (e.g. GCN, SAGE, or others). Algorithm 2 shows pseudo-code for the vanilla GCN. Finally, Algorithm 3 defines our full Network of GCN model (N-GCN) by plugging Algorithm 2 into Algorithm 1. Similarly, we list the algorithms for SAGE and Network of SAGE (N-SAGE) in the Appendix.

We can recover the original algorithms GCN (Kipf & Welling, 2017) and SAGE (Hamilton et al., 2017), respectively, by using Algorithms 3 (N-GCN) and 5 (N-SAGE, listed in Appendix) with $r = 1$, $K = 1$, identity CLASSIFIERFN, and modifying line 2 in Algorithm 1 to $P \leftarrow \hat{A}$. Moreover, we can recover original DCNN (Atwood & Towsley, 2016) by calling Algorithm 3 with $L = 1$, $r = 1$, modifying line 3 to $\hat{A} \leftarrow D^{-1}A$, and keeping $K > 1$ as their proposed model operates on the power series of the transition matrix i.e. *unmodified* random walks, like ours.

---

**Algorithm 1** General Implementation: Network of Graph Models

---

**Require:** $\hat{A}$ is a normalization of $A$
 1: **function** NETWORK(GRAPHMODELFN, $\hat{A}$, $X$, $L$, $r = 4$, $K = 6$, CLASSIFIERFN=FCLAYER)
 2:     $P \leftarrow I$
 3:     GraphModels $\leftarrow$ []
 4:     **for** $k = 1$ to $K$ **do**
 5:         **for** $i = 1$ to $r$ **do**
 6:             GraphModels.append(GRAPHMODELFN($P, X, L$))
 7:         $P \leftarrow \hat{A}P$
 8:     **return** CLASSIFIERFN(GraphModels)

---

**Algorithm 2** GCN (Kipf & Welling, 2017)

---

**Require:** $\hat{A}$ is a normalization of $A$
 1: **function** GCNMODEL($\hat{A}, X, L$)
 2:     $Z \leftarrow X$
 3:     **for** $i = 1$ to $L$ **do**
 4:         $Z \leftarrow \sigma(\hat{A}ZW^{(i)})$
 5:     **return** $Z$

---

**Algorithm 3** N-GCN

---

 1: **function** NGCN($A$, $X$, $L = 2$)
 2:     $D \leftarrow \mathbf{diag}(A\mathbf{1})$         ▷ Sum rows
 3:     $\hat{A} \leftarrow D^{-1/2}AD^{-1/2}$
 4:     **return** NETWORK(GCNMODEL, $\hat{A}, X, L$)

---

## 4 EXPERIMENTS

We follow the experimental setup by Kipf & Welling (2017) and Yang et al. (2016), including the provided dataset splits (train, validation, test) produced by Yang et al. (2016).

### 4.1 DATASETS

We experiment on three citation graph datasets: Pubmed, Citeseer, Cora, and a biological graph: Protein-Protein Interactions (PPI). We choose the aforementioned datasets because they are available online and are used by our baselines. The citation datasets are prepared by Yang et al. (2016), and the PPI dataset is prepared by Hamilton et al. (2017). Table 1 summarizes dataset statistics.

Each node in the citation datasets represents an article published in the corresponding journal. An edge between two nodes represents a citation from one article to another, and a label represents the subject of the article. Each dataset contains a binary Bag-of-Words (BoW) feature vector for each node. The BoW are extracted from the article abstract. Therefore, the task is to predict the subject of articles, given the BoW of their abstract and the citations to other (possibly labeled) articles. Following Yang et al. (2016) and Kipf & Welling (2017), we use 20 nodes per class for training, 500 (overall) nodes for validation, and 1000 nodes for evaluation. We note that the validation set is larger than training $|\mathcal{V}_L|$ for these datasets!

The PPI graph, as processed and described by Hamilton et al. (2017), consists of 24 disjoint subgraphs, each corresponding to a different human tissue. 20 of those subgraphs are used for training, 2 for validation, and 2 for testing, as partitioned by Hamilton et al. (2017).

### 4.2 BASELINE METHODS

For the citation datasets, we copy baseline numbers from Kipf & Welling (2017). These include label propagation (LP, Zhu et al. (2003)); semi-supervised embedding (SemiEmb, Weston et al. (2012)); manifold regularization (ManiReg, Belkin et al. (2006b)); skip-gram graph embeddings (DeepWalk Perozzi et al., 2014); Iterative Classification Algorithm (ICA, Lu & Getoor, 2003); Planetoid (Yang et al., 2016); vanilla GCN (Kipf & Welling, 2017). For PPI, we copy baseline numbers from (Hamilton et al., 2017), which include GraphSAGE with LSTM aggregation (SAGE-LSTM) and GraphSAGE with pooling aggregation (SAGE). Further, for all datasets, we use our implementation to run baselines DCNN (Atwood & Towsley, 2016), GCN (Kipf & Welling, 2017),

| Dataset | Type | Nodes $|\mathcal{V}|$ | Edges $|\mathcal{E}|$ | Classes $C$ | Features $F$ | Labeled nodes $|\mathcal{V}_L|$ |
|---------|------|-------|-------|---------|----------|---------------|
| Citeseer | citation | 3,327 | 4,732 | 6 (single class) | 3,703 | 120 |
| Cora | citation | 2,708 | 5,429 | 7 (single class) | 1,433 | 140 |
| Pubmed | citation | 19,717 | 44,338 | 3 (single class) | 500 | 60 |
| PPI | biological | 56,944 | 818,716 | 121 (multi-class) | 50 | 44,906 |

Table 1: Dataset used for experiments. For citation datasets, 20 training nodes per class are observed, with $|\mathcal{V}_L| = 20 \times C$

| | Method | Citeseer | Cora | Pubmed | PPI |
|-----|--------|----------|------|--------|-----|
| (a) | ManiReg (Belkin et al., 2006b) | 60.1 | 59.5 | 70.7 | – |
| (b) | SemiEmb (Weston et al., 2012) | 59.6 | 59.0 | 71.1 | – |
| (c) | LP (Zhu et al., 2003) | 45.3 | 68.0 | 63.0 | – |
| (d) | DeepWalk (Perozzi et al., 2014) | 43.2 | 67.2 | 65.3 | – |
| (e) | ICA (Lu & Getoor, 2003) | 69.1 | 75.1 | 73.9 | – |
| (f) | Planetoid (Yang et al., 2016) | 64.7 | 75.7 | 77.2 | – |
| (g) | GCN (Kipf & Welling, 2017) | 70.3 | 81.5 | 79.0 | – |
| (h) | SAGE-LSTM (Hamilton et al., 2017) | – | – | – | 61.2 |
| (i) | SAGE (Hamilton et al., 2017) | – | – | – | 60.0 |
| (j) | DCNN (our implementation) | 71.1 | 81.3 | 79.3 | 44.0 |
| (k) | GCN (our implementation) | 71.2 | 81.0 | 78.8 | 46.2 |
| (l) | SAGE (our implementation) | 63.5 | 77.4 | 77.6 | 59.8 |
| (m) | N-GCN (ours) | **72.2** | **83.0** | **79.5** | 46.8 |
| (n) | N-SAGE (ours) | 71.0 | 81.8 | 79.4 | **65.0** |

Table 2: Node classification performance (% accuracy for the first three, citation datasets, and f1 micro-averaged for multiclass PPI), using data splits of Yang et al. (2016); Kipf & Welling (2017) and Hamilton et al. (2017). We report the test accuracy corresponding to the run with the highest validation accuracy. Results in rows (a) through (g) are copied from Kipf & Welling (2017), rows (h) and (i) from (Hamilton et al., 2017), and (j) through (l) are generated using our code since we can recover other algorithms as explained in Section 3.6. Rows (m) and (n) are our models. Entries with "–" indicate that authors from whom we copied results did not run on those datasets. Nonetheless, we run all datasets using our implementation of the most-competitive baselines.

and SAGE (with pooling aggregation, Hamilton et al., 2017), as these baselines can be recovered as special cases of our algorithm, as explained in Section 3.6.

## 4.3 IMPLEMENTATION

We use TensorFlow(Abadi et al., 2015) to implement our methods, which we use to also measure the performance of baselines GCN, SAGE, and DCNN. For our methods and baselines, all GCN and SAGE modules that we train are 2 layers, where the first outputs 16 dimensions per node and the second outputs the number of classes (dataset-dependent). DCNN baseline has one layer and outputs 16 dimensions per node, and its channels (one per transition matrix power) are concatenated into a fully-connected layer that outputs the number of classes. We use $50\%$ dropout and L2 regularization of $10^{-5}$ for all of the aforementioned models.

## 4.4 NODE CLASSIFICATION ACCURACY

Table 2 shows node classification accuracy results. We run 20 different random initializations for every model (baselines and ours), train using Adam optimizer (Ba & Kingma, 2015) with learning rate of 0.01 for 600 steps, capturing the model parameters at peak validation accuracy to avoid overfitting. For our models, we sweep our hyperparameters $r$, $K$, and choice of classification sub-network $\in \{\text{fc}, \text{a}\}$. For baselines and our models, we choose the model with the highest accuracy on validation set, and use it to record metrics on the test set in Table 2.

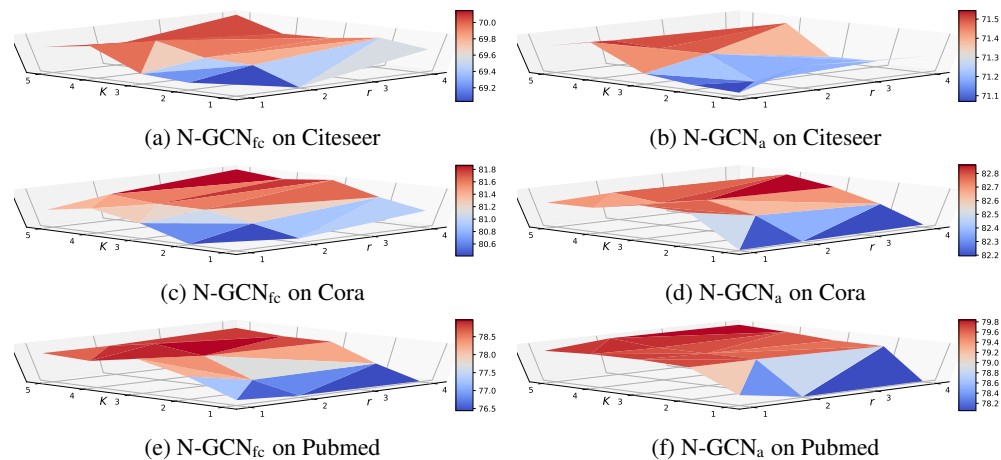

(a) N-GCN$_{fc}$ on Citeseer
(b) N-GCN$_a$ on Citeseer

(c) N-GCN$_{fc}$ on Cora
(d) N-GCN$_a$ on Cora

(e) N-GCN$_{fc}$ on Pubmed
(f) N-GCN$_a$ on Pubmed

Figure 2: Sensitivity Analysis. Model performance when varying random walk steps $K$ and replication factor $r$. Best viewed with zoom. Overall, model performance increases with larger values of $K$ and $r$. In addition, having random walk steps (larger $K$) boosts performance more than increasing model capacity (larger $r$), as seen by the cross-section cuts on along the $K$-axis versus the $r$-axis.

| Nodes per class | 5 | 10 | 20 | 100 |
|---|---|---|---|---|
| DCNN (our implementation) | $63.0 \pm 1.0$ | $72.3 \pm 0.4$ | $79.2 \pm 0.2$ | $82.6 \pm 0.3$ |
| GCN (our implementation) | $64.6 \pm 0.3$ | $70.0 \pm 3.7$ | $79.1 \pm 0.3$ | $81.8 \pm 0.3$ |
| SAGE (our implementation) | $69.0 \pm 1.4$ | $72.0 \pm 1.3$ | $77.2 \pm 0.5$ | $80.7 \pm 0.7$ |
| N-GCN$_a$ (ours) | $65.1 \pm 0.7$ | $71.2 \pm 1.1$ | $\mathbf{79.7} \pm 0.3$ | $\mathbf{83.0} \pm 0.4$ |
| N-GCN$_{fc}$ (ours) | $65.0 \pm 2.1$ | $71.7 \pm 0.7$ | $\mathbf{79.7} \pm 0.4$ | $82.9 \pm 0.3$ |
| N-SAGE$_a$ (ours) | $66.9 \pm 0.4$ | $73.4 \pm 0.7$ | $79.0 \pm 0.3$ | $82.5 \pm 0.2$ |
| N-SAGE$_{fc}$ (ours) | $\mathbf{70.7} \pm 0.4$ | $\mathbf{74.1} \pm 0.8$ | $78.5 \pm 1.0$ | $81.8 \pm 0.3$ |

Table 3: Node classification accuracy (in %) for our largest dataset (Pubmed) as we vary size of training data $\frac{|\mathcal{V}|}{C} \in \{5, 10, 20, 100\}$. We report mean and standard deviations on 10 runs. We use a different random seed for every run (i.e. selecting different labeled nodes), but the same 10 random seeds across models. Convolution-based methods (e.g. SAGE) work well with few training examples, but *unmodified* random walk methods (e.g. DCNN) work well with more training data. Our methods combine convolution and random walks, making them work well in both conditions.

Table 2 shows that N-GCN outperforms GCN (Kipf & Welling, 2017) and N-SAGE improves on SAGE for all datasets, showing that *unmodified* random walks indeed help in semi-supervised node classification. Finally, our proposed models acheive state-of-the-art on all datasets.

## 4.5 SENSITIVITY ANALYSIS

We analyze the impact of $K$ and $r$ on classification accuracy in Figure 2. We note that adding random walks by specifically setting $K > 1$ improves model accuracy due to the additional information, not due to increased model capacity. Contrast $K = 1, r > 1$ (i.e. mixture of GCNs, no random walks) with $K > 1, r = 1$ (i.e. N-GCN on random walks): in both scenarios, the model has more capacity, but the latter shows better performance. The same holds for SAGE, as shown in Appendix.

## 4.6 TOLERANCE TO FEATURE NOISE

We test our method under feature noise perturbations by removing node features at random. This is practical, as article authors might forget to include relevant terms in the article abstract, and more generally not all nodes will have the same amount of detailed information. Figure 3 shows that when features are removed, methods utilizing unmodified random walks: N-GCN, N-SAGE, and DCNN, outperform convolutional methods including GCN and SAGE. Moreover, the performance

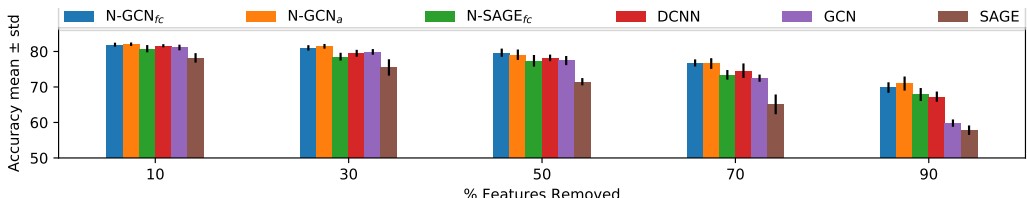

Figure 3: Classification accuracy for the Cora dataset with 20 labeled nodes per class ($|\mathcal{V}| = 20 \times C$), but features removed at random, averaging 10 runs. We use a different random seed for every run (i.e. removing different features per node), but the same 10 random seeds across models.

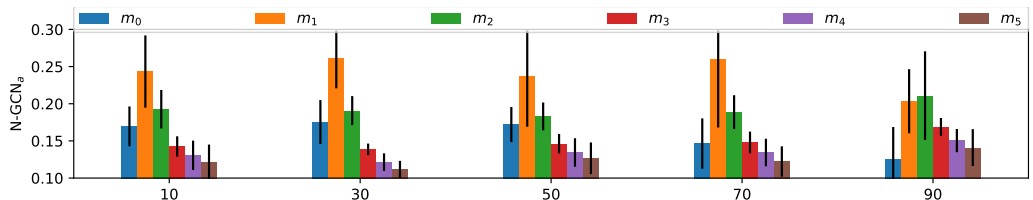

Figure 4: Attention weights ($m$) for N-GCN$_a$ when trained with feature removal perturbation on the Cora dataset. Removing features shifts the attention weights to the right, suggesting the model is relying more on long range dependencies.

gap widens as we remove more features. This suggests that our methods can somewhat recover removed features by *directly* pulling-in features from nearby and distant neighbors. We visualize in Figure 4 the attention weights as a function of % features removed. With little feature removal, there is some weight on $\hat{A}^0$, and the attention weights for $\hat{A}^1, \hat{A}^2, \ldots$ follow some decay function. Maliciously dropping features causes our model to shift its attention weights towards higher powers of $\hat{A}$.

## 5 RELATED WORK

The field of graph learning algorithms is quickly evolving. We review work most similar to ours.

Defferrard et al. (2016) define graph convolutions as a $K$-degree polynomial of the Laplacian, where the polynomial coefficients are learned. In their setup, the $K$-th degree Laplacian is a sparse square matrix where entry at $(i, j)$ will be zero if nodes $i$ and $j$ are more than $K$ hops apart. Their sparsity analysis also applies here. A minor difference is the adjacency normalization. We use $\hat{A}$ whereas they use the Laplacian defined as $I - \hat{A}$. Raising $\hat{A}$ to power $K$ will produce a square matrix with entry $(i, j)$ being the probability of random walker ending at node $i$ after $K$ steps from node $j$. The major difference is the order of random walk versus non-linearity. In particular, their model calculates learns a linear combination of $K$-degree polynomial and pass through classifier function $g$, as in $g(\sum_k q_k \widetilde{A}^k)$, while our (e.g. N-GCN) model calculates $\sum_k q_k g(\widetilde{A}^k)$, where $\widetilde{A}$ is $\hat{A}$ in our model and $I - \hat{A}$ in theirs, and our $g$ can be a GCN module. In fact, Defferrard et al. (2016) is also similar to work by Abu-El-Haija et al. (2017), as they both learn polynomial coefficients to some normalized adjacency matrix.

Atwood & Towsley (2016) propose DCNN, which calculates powers of the transition matrix and keeps each power in a separate channel until the classification sub-network at the end. Their model is therefore similar to our work in that it also falls under $\sum_k q_k g(\widetilde{A}^k)$. However, where their model multiplies features with each power $\widetilde{A}^k$ once, our model makes use of GCN's (Kipf & Welling, 2017) that multiply by $\widetilde{A}^k$ at every GCN layer (see Eq. 2). Thus, DCNN model (Atwood & Towsley, 2016) is a special case of ours, when GCN module contains only one layer, as explained in Section 3.6.

## 6 CONCLUSIONS AND FUTURE WORK

In this paper, we propose a meta-model that can run arbitrary Graph Convolution models, such as GCN (Kipf & Welling, 2017) and SAGE (Hamilton et al., 2017), on the output of random walks. Traditional Graph Convolution models operate on the normalized adjacency matrix. We make multiple instantiations of such models, feeding each instantiation a power of the adjacency matrix, and then concatenating the output of all instances into a classification sub-network. Our model, Network of GCNs (and similarly, Network of SAGE), is end-to-end trainable, and is able to directly learn information across near or distant neighbors. We inspect the distribution of parameter weights in our classification sub-network, which reveal to us that our model is effectively able to circumvent adversarial perturbations on the input by shifting weights towards model instances consuming higher powers of the adjacency matrix. For future work, we plan to extend our methods to a stochastic implementation and tackle other (larger) graph datasets.

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

# 7 APPENDIX

## 7.1 ALGORITHM FOR NETWORK OF SAGE

Algorithms 4 and 5, respectively, define SAGE Hamilton et al. (2017) and Network of SAGE (N-SAGE). Algorithm 4 assumes mean-pool aggregation by Hamilton et al. (2017), which performs on-par to their top performer max-pool aggregation. Further, Algorithm 4 operates in full-batch while Hamilton et al. (2017) offer a stochastic implementation with edge sampling. Nonetheless, their proposed stochastic implementation should be wrapped in a network, though we would need a way to approximate (e.g. sample entries) from dense $\hat{A}^k$ as $k$ increases. We leave this as future work.

---

**Algorithm 4** SAGE Model (Hamilton et al., 2017)

**Require:** $\hat{A}$ is a normalization of $A$
1: **function** SAGEMODEL($\hat{A}$, $X$, $L$)
2:     $Z \leftarrow X$
3:     **for** $i = 1$ to $L$ **do**
4:         $Z \leftarrow \sigma([\ Z \ \vdots \ \hat{A}Z\ ] W^{(i)})$
5:         $Z \leftarrow \text{L2NORMALIZEROWS}(Z)$
6:     **return** $Z$

**Algorithm 5** N-SAGE

1: **function** NSAGE($A$, $X$)
2:     $D \leftarrow \textbf{diag}(A\mathbf{1})$          ▷ Sum rows
3:     $\hat{A} \leftarrow D^{-1}A$
4:     **return** NETWORK(SAGEMODEL, $\hat{A}$, $X$, 2)

---

Using SAGE with mean-pooling aggregation is very similar to a vanilla GCN model but with three differences. First, the choice of adjacency normalization ($D^{-1}A$ versus $D^{-\frac{1}{2}}AD^{-\frac{1}{2}}$). Second, the skip connections in line 4, which concatenates the features with the adjacency-multiplied (i.e. diffused) features. We believe this is analogous in intuition of incorporating $\hat{A}^0$ in our model, which keeps the original features. Third, the use of node-wise L2 feature normalization at line 5, which is equivalent to applying a layernorm transformation J. Ba (2016). Nonetheless, it is worth noting Hamilton et al. (2017)'s formulation of SAGE is flexible to allow different aggregations, such as max-pooling or LSTM, which further deviates SAGE from GCN.

## 7.2 SENSITIVITY ANALYSIS

Earlier, in Table 2, we showed the test performance corresponding to the model performing best on the validation split. The number of labeled nodes are small, and such model selection is important to avoid overfitting. For example, there can be up to $10\%$ relative test accuracy difference when training the *same* model architecture but with *different* random seed. In this section, we programatically sweep hyperparameters $r$, $K$, choice of classification network ($\in \{\text{fc}, \text{a}\}$), and whether or not we enable $\hat{A}^0$, for both N-GCN and N-SAGE models.

The settings when ($K = 1$, $r = 1$, and $\hat{A}^0$ disabled), correspond to the vanilla base model. Further, the settings when ($K = 1$, $r > 1$, and $\hat{A}^0$ disabled), correspond to an ensemble of the base model. These cases are outperformed when $K > 1$, showing that *unmodified* random walks indeed help these convolutional methods perform better, by gathering information from nearby and distant nodes.

The automatically generated tables are shown below:

|  | $K = 1$ | $K = 2$ | $K = 3$ | $K = 4$ | $K = 5$ |
|---|---|---|---|---|---|
| $r = 1$ | $79.0 \pm 0.163$ | $\mathbf{79.5 \pm 0.100}$ | $79.3 \pm 0.372$ | $79.4 \pm 0.234$ | $79.4 \pm 0.337$ |
| $r = 2$ | $79.1 \pm 0.283$ | $79.3 \pm 0.241$ | $79.4 \pm 0.134$ | $79.4 \pm 0.146$ | $79.4 \pm 0.160$ |
| $r = 4$ | $78.9 \pm 0.181$ | $79.4 \pm 0.161$ | $79.3 \pm 0.163$ | $\mathbf{79.5 \pm 0.302}$ | $\mathbf{79.5 \pm 0.227}$ |

Table 4: N-GCN$_\text{a}$ results on Citeseer dataset, with $\hat{A}^0$ disabled. Top-left entry corresponds to vanilla GCN. Left column corresponds to ensemble of GCN models.

|       | $K = 1$ | $K = 2$ | $K = 3$ | $K = 4$ | $K = 5$ |
|-------|---------|---------|---------|---------|---------|
| $r = 1$ | $78.1 \pm 0.339$ | $79.6 \pm 0.293$ | $79.8 \pm 0.189$ | $79.7 \pm 0.170$ | $79.6 \pm 0.243$ |
| $r = 2$ | $77.3 \pm 0.125$ | $79.7 \pm 0.171$ | $79.6 \pm 0.189$ | $79.6 \pm 0.138$ | $\mathbf{79.9} \pm 0.177$ |
| $r = 4$ | $77.3 \pm 0.287$ | $79.5 \pm 0.396$ | $79.5 \pm 0.219$ | $79.7 \pm 0.149$ | $\mathbf{79.9} \pm 0.189$ |

Table 5: N-GCN$_a$ results on Citeseer dataset, with $\hat{A}^0$ enabled.

|       | $K = 1$ | $K = 2$ | $K = 3$ | $K = 4$ | $K = 5$ |
|-------|---------|---------|---------|---------|---------|
| $r = 1$ | $-$ | $78.6 \pm 0.723$ | $78.7 \pm 0.407$ | $78.7 \pm 0.530$ | $78.0 \pm 0.690$ |
| $r = 2$ | $78.5 \pm 0.353$ | $77.9 \pm 0.234$ | $78.5 \pm 0.724$ | $78.8 \pm 0.562$ | $\mathbf{79.1} \pm 0.267$ |
| $r = 4$ | $78.4 \pm 0.499$ | $78.4 \pm 0.716$ | $78.9 \pm 0.306$ | $78.9 \pm 0.385$ | $79.0 \pm 0.228$ |

Table 6: N-GCN$_{fc}$ results on Citeseer dataset, with $\hat{A}^0$ disabled. Left column corresponds to ensemble of GCN models.

|       | $K = 1$ | $K = 2$ | $K = 3$ | $K = 4$ | $K = 5$ |
|-------|---------|---------|---------|---------|---------|
| $r = 1$ | $76.5 \pm 1.490$ | $78.2 \pm 1.290$ | $79.2 \pm 1.061$ | $78.5 \pm 0.963$ | $78.7 \pm 1.384$ |
| $r = 2$ | $76.1 \pm 1.118$ | $77.1 \pm 1.152$ | $78.8 \pm 1.479$ | $\mathbf{79.4} \pm 0.754$ | $78.7 \pm 0.612$ |
| $r = 4$ | $76.0 \pm 0.770$ | $77.2 \pm 0.785$ | $78.7 \pm 0.716$ | $78.7 \pm 0.953$ | $79.0 \pm 0.313$ |

Table 7: N-GCN$_{fc}$ results on Citeseer dataset, with $\hat{A}^0$ enabled.

|       | $K = 1$ | $K = 2$ | $K = 3$ | $K = 4$ | $K = 5$ |
|-------|---------|---------|---------|---------|---------|
| $r = 1$ | $76.0 \pm 1.239$ | $77.0 \pm 0.856$ | $77.3 \pm 0.682$ | $77.4 \pm 0.419$ | $77.3 \pm 0.979$ |
| $r = 2$ | $76.4 \pm 1.219$ | $77.6 \pm 0.508$ | $77.6 \pm 0.414$ | $77.7 \pm 0.586$ | $\mathbf{78.0} \pm 0.250$ |
| $r = 4$ | $76.5 \pm 0.863$ | $77.3 \pm 0.198$ | $77.8 \pm 0.525$ | $77.9 \pm 0.522$ | $77.6 \pm 0.393$ |

Table 8: N-SAGE$_a$ results on Citeseer dataset, with $\hat{A}^0$ disabled. Top-left entry corresponds to vanilla SAGE. Left column corresponds to ensemble of SAGE models.

|       | $K = 1$ | $K = 2$ | $K = 3$ | $K = 4$ | $K = 5$ |
|-------|---------|---------|---------|---------|---------|
| $r = 1$ | $73.4 \pm 1.264$ | $76.1 \pm 0.306$ | $76.8 \pm 0.647$ | $76.6 \pm 0.623$ | $77.0 \pm 0.340$ |
| $r = 2$ | $75.2 \pm 0.597$ | $76.0 \pm 0.453$ | $76.4 \pm 0.241$ | $77.2 \pm 0.306$ | $77.3 \pm 0.869$ |
| $r = 4$ | $74.9 \pm 0.530$ | $76.8 \pm 0.535$ | $77.0 \pm 0.289$ | $\mathbf{77.5} \pm 0.407$ | $77.3 \pm 0.318$ |

Table 9: N-SAGE$_a$ results on Citeseer dataset, with $\hat{A}^0$ enabled.

|       | $K = 1$ | $K = 2$ | $K = 3$ | $K = 4$ | $K = 5$ |
|-------|---------|---------|---------|---------|---------|
| $r = 1$ | $-$ | $76.3 \pm 1.545$ | $76.7 \pm 1.098$ | $78.0 \pm 1.427$ | $77.3 \pm 1.038$ |
| $r = 2$ | $76.6 \pm 1.196$ | $77.3 \pm 1.309$ | $77.8 \pm 0.746$ | $77.5 \pm 0.836$ | $77.5 \pm 0.298$ |
| $r = 4$ | $76.5 \pm 0.602$ | $\mathbf{78.1} \pm 1.239$ | $77.6 \pm 0.287$ | $76.9 \pm 0.472$ | $77.7 \pm 1.119$ |

Table 10: N-SAGE$_{fc}$ results on Citeseer dataset, with $\hat{A}^0$ disabled. Left column corresponds to ensemble of SAGE models.

|       | $K = 1$ | $K = 2$ | $K = 3$ | $K = 4$ | $K = 5$ |
|-------|---------|---------|---------|---------|---------|
| $r = 1$ | $72.9 \pm 0.972$ | $75.9 \pm 0.922$ | $75.5 \pm 0.499$ | $76.6 \pm 1.641$ | $76.8 \pm 0.589$ |
| $r = 2$ | $75.3 \pm 0.879$ | $76.1 \pm 1.237$ | $76.6 \pm 0.579$ | $76.4 \pm 0.383$ | $76.2 \pm 0.626$ |
| $r = 4$ | $75.3 \pm 1.730$ | $76.4 \pm 1.186$ | $76.6 \pm 0.576$ | $76.8 \pm 0.450$ | $\mathbf{77.4} \pm 0.712$ |

Table 11: N-SAGE$_{fc}$ results on Citeseer dataset, with $\hat{A}^0$ enabled.

|       | $K = 1$ | $K = 2$ | $K = 3$ | $K = 4$ | $K = 5$ |
|-------|---------|---------|---------|---------|---------|
| $r = 1$ | $79.0 \pm 0.163$ | $\mathbf{79.5} \pm 0.100$ | $79.3 \pm 0.372$ | $79.4 \pm 0.234$ | $79.4 \pm 0.337$ |
| $r = 2$ | $79.1 \pm 0.283$ | $79.3 \pm 0.241$ | $79.4 \pm 0.134$ | $79.4 \pm 0.146$ | $79.4 \pm 0.160$ |
| $r = 4$ | $78.9 \pm 0.181$ | $79.4 \pm 0.161$ | $79.3 \pm 0.163$ | $\mathbf{79.5} \pm 0.302$ | $\mathbf{79.5} \pm 0.227$ |

Table 12: N-GCN$_a$ results on Cora dataset, with $\hat{A}^0$ disabled. Top-left entry corresponds to vanilla GCN. Left column corresponds to ensemble of GCN models.

|       | $K = 1$ | $K = 2$ | $K = 3$ | $K = 4$ | $K = 5$ |
|-------|---------|---------|---------|---------|---------|
| $r = 1$ | $78.1 \pm 0.339$ | $79.6 \pm 0.293$ | $79.8 \pm 0.189$ | $79.7 \pm 0.170$ | $79.6 \pm 0.243$ |
| $r = 2$ | $77.3 \pm 0.125$ | $79.7 \pm 0.171$ | $79.6 \pm 0.189$ | $79.6 \pm 0.138$ | $\mathbf{79.9} \pm 0.177$ |
| $r = 4$ | $77.3 \pm 0.287$ | $79.5 \pm 0.396$ | $79.5 \pm 0.219$ | $79.7 \pm 0.149$ | $\mathbf{79.9} \pm 0.189$ |

Table 13: N-GCN$_a$ results on Cora dataset, with $\hat{A}^0$ enabled.

|       | $K = 1$ | $K = 2$ | $K = 3$ | $K = 4$ | $K = 5$ |
|-------|---------|---------|---------|---------|---------|
| $r = 1$ | $-$ | $78.6 \pm 0.723$ | $78.7 \pm 0.407$ | $78.7 \pm 0.530$ | $78.0 \pm 0.690$ |
| $r = 2$ | $78.5 \pm 0.353$ | $77.9 \pm 0.234$ | $78.5 \pm 0.724$ | $78.8 \pm 0.562$ | $\mathbf{79.1} \pm 0.267$ |
| $r = 4$ | $78.4 \pm 0.499$ | $78.4 \pm 0.716$ | $78.9 \pm 0.306$ | $78.9 \pm 0.385$ | $79.0 \pm 0.228$ |

Table 14: N-GCN$_{fc}$ results on Cora dataset, with $\hat{A}^0$ disabled. Left column corresponds to ensemble of GCN models.

|       | $K = 1$ | $K = 2$ | $K = 3$ | $K = 4$ | $K = 5$ |
|-------|---------|---------|---------|---------|---------|
| $r = 1$ | $76.5 \pm 1.490$ | $78.2 \pm 1.290$ | $79.2 \pm 1.061$ | $78.5 \pm 0.963$ | $78.7 \pm 1.384$ |
| $r = 2$ | $76.1 \pm 1.118$ | $77.1 \pm 1.152$ | $78.8 \pm 1.479$ | $\mathbf{79.4} \pm 0.754$ | $78.7 \pm 0.612$ |
| $r = 4$ | $76.0 \pm 0.770$ | $77.2 \pm 0.785$ | $78.7 \pm 0.716$ | $78.7 \pm 0.953$ | $79.0 \pm 0.313$ |

Table 15: N-GCN$_{fc}$ results on Cora dataset, with $\hat{A}^0$ enabled.

|       | $K = 1$ | $K = 2$ | $K = 3$ | $K = 4$ | $K = 5$ |
|-------|---------|---------|---------|---------|---------|
| $r = 1$ | $76.0 \pm 1.239$ | $77.0 \pm 0.856$ | $77.3 \pm 0.682$ | $77.4 \pm 0.419$ | $77.3 \pm 0.979$ |
| $r = 2$ | $76.4 \pm 1.219$ | $77.6 \pm 0.508$ | $77.6 \pm 0.414$ | $77.7 \pm 0.586$ | $\mathbf{78.0} \pm 0.250$ |
| $r = 4$ | $76.5 \pm 0.863$ | $77.3 \pm 0.198$ | $77.8 \pm 0.525$ | $77.9 \pm 0.522$ | $77.6 \pm 0.393$ |

Table 16: N-SAGE$_a$ results on Cora dataset, with $\hat{A}^0$ disabled. Top-left entry corresponds to vanilla SAGE. Left column corresponds to ensemble of SAGE models.

|       | $K = 1$ | $K = 2$ | $K = 3$ | $K = 4$ | $K = 5$ |
|-------|---------|---------|---------|---------|---------|
| $r = 1$ | $73.4 \pm 1.264$ | $76.1 \pm 0.306$ | $76.8 \pm 0.647$ | $76.6 \pm 0.623$ | $77.0 \pm 0.340$ |
| $r = 2$ | $75.2 \pm 0.597$ | $76.0 \pm 0.453$ | $76.4 \pm 0.241$ | $77.2 \pm 0.306$ | $77.3 \pm 0.869$ |
| $r = 4$ | $74.9 \pm 0.530$ | $76.8 \pm 0.535$ | $77.0 \pm 0.289$ | $\mathbf{77.5} \pm 0.407$ | $77.3 \pm 0.318$ |

Table 17: N-SAGE$_a$ results on Cora dataset, with $\hat{A}^0$ enabled.

|       | $K = 1$ | $K = 2$ | $K = 3$ | $K = 4$ | $K = 5$ |
|-------|---------|---------|---------|---------|---------|
| $r = 1$ | $-$ | $76.3 \pm 1.545$ | $76.7 \pm 1.098$ | $78.0 \pm 1.427$ | $77.3 \pm 1.038$ |
| $r = 2$ | $76.6 \pm 1.196$ | $77.3 \pm 1.309$ | $77.8 \pm 0.746$ | $77.5 \pm 0.836$ | $77.5 \pm 0.298$ |
| $r = 4$ | $76.5 \pm 0.602$ | $\mathbf{78.1} \pm 1.239$ | $77.6 \pm 0.287$ | $76.9 \pm 0.472$ | $77.7 \pm 1.119$ |

Table 18: N-SAGE$_{fc}$ results on Cora dataset, with $\hat{A}^0$ disabled. Left column corresponds to ensemble of SAGE models.

|       | $K = 1$ | $K = 2$ | $K = 3$ | $K = 4$ | $K = 5$ |
|-------|---------|---------|---------|---------|---------|
| $r = 1$ | $72.9 \pm 0.972$ | $75.9 \pm 0.922$ | $75.5 \pm 0.499$ | $76.6 \pm 1.641$ | $76.8 \pm 0.589$ |
| $r = 2$ | $75.3 \pm 0.879$ | $76.1 \pm 1.237$ | $76.6 \pm 0.579$ | $76.4 \pm 0.383$ | $76.2 \pm 0.626$ |
| $r = 4$ | $75.3 \pm 1.730$ | $76.4 \pm 1.186$ | $76.6 \pm 0.576$ | $76.8 \pm 0.450$ | $\mathbf{77.4} \pm 0.712$ |

Table 19: N-SAGE$_{fc}$ results on Cora dataset, with $\hat{A}^0$ enabled.

|       | $K = 1$ | $K = 2$ | $K = 3$ | $K = 4$ | $K = 5$ |
|-------|---------|---------|---------|---------|---------|
| $r = 1$ | $79.0 \pm 0.163$ | $\mathbf{79.5} \pm 0.100$ | $79.3 \pm 0.372$ | $79.4 \pm 0.234$ | $79.4 \pm 0.337$ |
| $r = 2$ | $79.1 \pm 0.283$ | $79.3 \pm 0.241$ | $79.4 \pm 0.134$ | $79.4 \pm 0.146$ | $79.4 \pm 0.160$ |
| $r = 4$ | $78.9 \pm 0.181$ | $79.4 \pm 0.161$ | $79.3 \pm 0.163$ | $\mathbf{79.5} \pm 0.302$ | $\mathbf{79.5} \pm 0.227$ |

Table 20: N-GCN$_a$ results on Pubmed dataset, with $\hat{A}^0$ disabled. Top-left entry corresponds to vanilla GCN. Left column corresponds to ensemble of GCN models.

|       | $K = 1$ | $K = 2$ | $K = 3$ | $K = 4$ | $K = 5$ |
|-------|---------|---------|---------|---------|---------|
| $r = 1$ | $78.1 \pm 0.339$ | $79.6 \pm 0.293$ | $79.8 \pm 0.189$ | $79.7 \pm 0.170$ | $79.6 \pm 0.243$ |
| $r = 2$ | $77.3 \pm 0.125$ | $79.7 \pm 0.171$ | $79.6 \pm 0.189$ | $79.6 \pm 0.138$ | $\mathbf{79.9} \pm 0.177$ |
| $r = 4$ | $77.3 \pm 0.287$ | $79.5 \pm 0.396$ | $79.5 \pm 0.219$ | $79.7 \pm 0.149$ | $\mathbf{79.9} \pm 0.189$ |

Table 21: N-GCN$_\text{a}$ results on Pubmed dataset, with $\hat{A}^0$ enabled.

|       | $K = 1$ | $K = 2$ | $K = 3$ | $K = 4$ | $K = 5$ |
|-------|---------|---------|---------|---------|---------|
| $r = 1$ | $-$ | $78.6 \pm 0.723$ | $78.7 \pm 0.407$ | $78.7 \pm 0.530$ | $78.0 \pm 0.690$ |
| $r = 2$ | $78.5 \pm 0.353$ | $77.9 \pm 0.234$ | $78.5 \pm 0.724$ | $78.8 \pm 0.562$ | $\mathbf{79.1} \pm 0.267$ |
| $r = 4$ | $78.4 \pm 0.499$ | $78.4 \pm 0.716$ | $78.9 \pm 0.306$ | $78.9 \pm 0.385$ | $79.0 \pm 0.228$ |

Table 22: N-GCN$_\text{fc}$ results on Pubmed dataset, with $\hat{A}^0$ disabled. Left column corresponds to ensemble of GCN models.

|       | $K = 1$ | $K = 2$ | $K = 3$ | $K = 4$ | $K = 5$ |
|-------|---------|---------|---------|---------|---------|
| $r = 1$ | $76.5 \pm 1.490$ | $78.2 \pm 1.290$ | $79.2 \pm 1.061$ | $78.5 \pm 0.963$ | $78.7 \pm 1.384$ |
| $r = 2$ | $76.1 \pm 1.118$ | $77.1 \pm 1.152$ | $78.8 \pm 1.479$ | $\mathbf{79.4} \pm 0.754$ | $78.7 \pm 0.612$ |
| $r = 4$ | $76.0 \pm 0.770$ | $77.2 \pm 0.785$ | $78.7 \pm 0.716$ | $78.7 \pm 0.953$ | $79.0 \pm 0.313$ |

Table 23: N-GCN$_\text{fc}$ results on Pubmed dataset, with $\hat{A}^0$ enabled.

|       | $K = 1$ | $K = 2$ | $K = 3$ | $K = 4$ | $K = 5$ |
|-------|---------|---------|---------|---------|---------|
| $r = 1$ | $76.0 \pm 1.239$ | $77.0 \pm 0.856$ | $77.3 \pm 0.682$ | $77.4 \pm 0.419$ | $77.3 \pm 0.979$ |
| $r = 2$ | $76.4 \pm 1.219$ | $77.6 \pm 0.508$ | $77.6 \pm 0.414$ | $77.7 \pm 0.586$ | $\mathbf{78.0} \pm 0.250$ |
| $r = 4$ | $76.5 \pm 0.863$ | $77.3 \pm 0.198$ | $77.8 \pm 0.525$ | $77.9 \pm 0.522$ | $77.6 \pm 0.393$ |

Table 24: N-SAGE$_\text{a}$ results on Pubmed dataset, with $\hat{A}^0$ disabled. Top-left entry corresponds to vanilla SAGE. Left column corresponds to ensemble of SAGE models.

|       | $K = 1$ | $K = 2$ | $K = 3$ | $K = 4$ | $K = 5$ |
|-------|---------|---------|---------|---------|---------|
| $r = 1$ | $73.4 \pm 1.264$ | $76.1 \pm 0.306$ | $76.8 \pm 0.647$ | $76.6 \pm 0.623$ | $77.0 \pm 0.340$ |
| $r = 2$ | $75.2 \pm 0.597$ | $76.0 \pm 0.453$ | $76.4 \pm 0.241$ | $77.2 \pm 0.306$ | $77.3 \pm 0.869$ |
| $r = 4$ | $74.9 \pm 0.530$ | $76.8 \pm 0.535$ | $77.0 \pm 0.289$ | $\mathbf{77.5} \pm 0.407$ | $77.3 \pm 0.318$ |

Table 25: N-SAGE$_\text{a}$ results on Pubmed dataset, with $\hat{A}^0$ enabled.

|       | $K = 1$ | $K = 2$ | $K = 3$ | $K = 4$ | $K = 5$ |
|-------|---------|---------|---------|---------|---------|
| $r = 1$ | $-$ | $76.3 \pm 1.545$ | $76.7 \pm 1.098$ | $78.0 \pm 1.427$ | $77.3 \pm 1.038$ |
| $r = 2$ | $76.6 \pm 1.196$ | $77.3 \pm 1.309$ | $77.8 \pm 0.746$ | $77.5 \pm 0.836$ | $77.5 \pm 0.298$ |
| $r = 4$ | $76.5 \pm 0.602$ | $\mathbf{78.1} \pm 1.239$ | $77.6 \pm 0.287$ | $76.9 \pm 0.472$ | $77.7 \pm 1.119$ |

Table 26: N-SAGE$_\text{fc}$ results on Pubmed dataset, with $\hat{A}^0$ disabled. Left column corresponds to ensemble of SAGE models.

|       | $K = 1$ | $K = 2$ | $K = 3$ | $K = 4$ | $K = 5$ |
|-------|---------|---------|---------|---------|---------|
| $r = 1$ | $72.9 \pm 0.972$ | $75.9 \pm 0.922$ | $75.5 \pm 0.499$ | $76.6 \pm 1.641$ | $76.8 \pm 0.589$ |
| $r = 2$ | $75.3 \pm 0.879$ | $76.1 \pm 1.237$ | $76.6 \pm 0.579$ | $76.4 \pm 0.383$ | $76.2 \pm 0.626$ |
| $r = 4$ | $75.3 \pm 1.730$ | $76.4 \pm 1.186$ | $76.6 \pm 0.576$ | $76.8 \pm 0.450$ | $\mathbf{77.4} \pm 0.712$ |

Table 27: N-SAGE$_\text{fc}$ results on Pubmed dataset, with $\hat{A}^0$ enabled.

