# OpenReview forum: "Network of Graph Convolutional Networks Trained on Random Walks"
_ICLR.cc/2018/Conference — Reject_

### Official Review · AnonReviewer2 · 2017-11-25
**possibly interesting ideas**

**Rating:** 5
**Confidence:** 2

**Review:**

The paper proposes a novel graph convolutional network in which a variety of random walk steps are involved with multiple GCNs.

The basic idea, introducing long rage dependecy, would be interesting. Robustness for the feature remove is also interesting.

The validation set would be important for the proposed method, but for creating larger validation set, labeled training set would become small. How the good balance of training-and-validation can be determined?

Discussing choice of the degree would be informative. In introducing many degrees (GCNs) for small labeled nodes semi-supervised setting seems to cause over-fitting.

---

> ### Author Response · Authors · 2017-12-24
> **Data splits are not ours. Added experiments with Mixture of GCNs to give it more capacity .**
>
> Thank you for taking the time to review our work!
>
> * Balance on training and validation:
>
> We re-use the splits created by Planetoid paper (including train, validate, test) and we do not control it in this paper.
>
>
> * Degree of GCNs:
>
> We assume that you meant the "capacity" (e.g. number of parameters) of GCNs. We now conduct experiments in Appendix on GCNs when we give them more parameters, and we show that they perform worse than our models, showing that our methods are out-performing because of random walks, and not necessarily more parameters.

---

### Official Review · AnonReviewer3 · 2017-11-26
**Interesting approach, but lacks justification**

**Rating:** 5
**Confidence:** 4

**Review:**

In this work a new network of GCNs is proposed. Different GCNs utilize different powers of the transition matrix to capture varying neighborhoods in a graph. As an aggregation mechanism of the GCN modules two approaches are considered: a fully connected layer on top of stacked features and attention mechanism that uses a scalar weight per GCN. The later allows for better interpretability of the effects of varying degree of neighborhoods in a graph.

Proposed approach, as authors noted themselves, is quite similar to DCNN (Atwood and Towsley, 2016) and becomes equivalent if the combined GCNs have one layer each. While comparison to vanilla GCN is quite extensive, there is no comparison to DCNN at all. I would be curious to see at least portion of the experiments of the DCNN paper with the proposed approach, where the importance of number of GCN layers is addressed. DCNN did well on Cora and Pubmed when more training samples were used. It also was tested on graph classification datasets, but the results were not as good for some of the datasets. I think that comparison to DCNN is important to justify the importance of using multilayer GCN modules.

Some questions and concerns:
- I could not quite figure out how many layers did each GCN have in the experiments and how impactful is this parameter
- Why is it necessary to replicate GCNs for each of the transition matrix powers? In section 4.3 it is mentioned that replication factors r = 1 and r = 4 were used, but it is not clear from Table 2 what are the results for respective r.
- Early stopping implementation seems a bit too intense. "We invoke many runs over all datasets" - how many? Mean and standard deviation are reported for top 3 performers, which is not enough to get a sense of standard deviation and mean. Kipf and Welling (2017) report results over 100 runs without selecting top performers if I understood correctly their setup. Could you please report mean and standard deviation of all the runs? Given relatively small performance improvement (comparatively to GCN), more than 3 (selected) runs are needed for comparison.
- I liked the attention idea and its interpretation in Fig. 2. Could you please add the error bars for the attention weights. It is interesting to see them shifting towards higher powers of the transition matrix, but also it is important to know if this phenomena is statistically significant.
- Following up on the previous item - did you try not including self connections when computing transition matrix powers? This way the effect of different degrees of neighborhoods in a graph could be understood better. When self-connections are present, each subsequent transition matrix power contains neighborhoods of lower degrees and interpretation becomes not as apparent.

Minor comments:
- Understanding of this paper quite heavily relies on the reader knowing Kipf and Welling (2017) paper. Particularly, the comment about approximations derived by Kipf and Welling (2017) in Section 3.3 and how directed graph was converted to undirected (Section 4.1) require a bit more details.
- I am not quite sure why Section 2.3 is needed. Connection to graph embeddings is not given much attention in the paper later on (except t-SNE picture).
- Typo in Fig. 1 caption - right and left are mixed up.
- Typo in footnote on page 3.

---

> ### Author Response · Authors · 2017-12-24
> **Added DCNN experiments**
>
> Thank you for your review! It made our work much better!
>
> * Compare with DCNN:
> We added experiments to DCNN, and clearly explained how they are a special case of ours in new Section 3.6. We outperform them in the "standard" setup that was used by Kipf and Planetoid (i.e. 20 nodes per class). However, DCNN is showing more power than GCN's with more training data (e.g. see table on Pubmed, up to 100 labeled nodes).
>
>
> * Layers of GCN?
> Thanks! We now made it clear in writing. Our GCN and SAGE modules for both our models and baselines use 2 layers.
>
>
> * Why replication r > 1?
> We added extensive evaluation in Appendix. More "r" helps, similar to "ensemble of classifiers" (e.g. mixture of experts). This seems to help on validation+test but not on train accuracy, as all models reach ~100% accuracy on training anyway.
>
>
> * Early stopping and "many runs" for validation.
> We beleive that model selection we do is acceptable. We choose models based on *validation* accuracy. In fact, we now choose the top 1 model based on validation accuracy and report its test accuracy, which is the true practical setting. We do many runs (total == thousands, for all parameter sweeps) and put mean and standard deviation in appendix. Also, we now re-ran all experiments without early stopping.
>
>
> * Add error bars to attention.
> Good idea. Now done. Thank you!
>
>
> * Self-connections:
> We add self connections, and already mentioned it in at least 2 places as we follow Kipf's setup.
>
>
> * Understanding paper requires knowing Kipf's work
> We tried to explain what it means that approximations "still valid". Is it better now? we tried our best to make the paper stand on its own and will continue doing so before the camera ready (in hopes it gets accepted)
>
>
> * Section 2.3 not needed.
> We feel that it gives the reader background of embeddings on adjacency VS embeddings using random walks. It also gives us defines \mathcal{T} in terms of D and A (which might be known by many readers and they could skim that section).
>
>
> * Typos:
> We fixed them. Thank you for pointing them out.

---

> ### Author Response · Authors · 2018-01-06
> **Added PPI dataset (used by GraphSAGE)**
>
> We added experiments for PPI dataset (we downloaded it from the GraphSAGE paper).
>
> from SAGE authors:
> SAGE-LSTM gets 61.2
> SAGE [i.e. pooling] gets 60.0
>
> Our implementation of SAGE gets 59.8
> Our method (N-SAGE) gets 65.0
>
> Results are added to the table.

---

### Official Review · AnonReviewer1 · 2017-11-27
**Interesting idea to boost GCN.**

**Rating:** 6
**Confidence:** 5

**Review:**

The paper presents a Network of Graph Convolutional Networks (NGCNs) that uses
random walk statistics to extract information from near and distant neighbors
in the graph.

The authors show that a 2-layer Graph Convolutional Network, with linear
activation and W0 as identity matrix, reduces to a one-step random walk.
They build on this notion to  introduce the idea to make the GCN directly operate
on random walk statistics to better model information across distant nodes.

Given that it is not clear how many steps of random walk to use a-priori it is
proposed to make a mixture of models whose outputs are combined by a
softmax classifier, or by an attention based mixing (learning the mixing coefficients).

I find that the comparison can be considered slightly unfair as NGCN has k-times
the number of GCN models in it. Did the authors compare with a deeper GCN, or
simply with a mixture of plain GCN using one-step random walk?
The datasets used for comparison are extremely simple, and I am glad that the
authors point out that this is a significant issue for benchmark driven research.
However, doing calibration on a subset of the validation nodes via gradient
descent is not very clean as by doing it one implicitly uses those nodes for training.
The improvement of the calibrated model on 5 nodes per class (Table 3) seems
to hint that this peeking into the validation is indeed happening.

The authors mention that feeding explicitly the information on distant nodes
makes learning easier and that otherwise such information it would be hard to
extract from stacking several GCN layers. While this is true for the small datasets
usually considered it is not clear at all whether this still holds when we will
have large scale graph benchmarks.

Experiments are well conducted but lack a comparison with GraphSAGE and MoNet,
which are the reference models for the selected benchmarks. A comparison would have made the contribution stronger in my opinion. Improvements in performance are minor
except for decimated inputs setting reported in Table 3. In this last case though
no statistics over multiple runs are shown.

Overall I like the interpretation, even if a bit forced, of GCN as using one-step
random walk statistics. The paper is clearly written.
The main issue I have with the approach is that it does not bring a very novel
way to perform deep learning on graphs, but rather improves marginally upon
a well established one.

---

> ### Author Response · Authors · 2017-12-24
> **Compared with Mixture of GCNs, added GraphSAGE, removed Calibration text and experiments**
>
> Thank you for your review! It made our work much better!
>
> * It is unfair to compare N-GCN which has k-times more parameters to GCN.
>
> We tried deeper GCN and the results were worse. We also tried >16 hidden dimensions (e.g. 32, 64, 512), and the results were also worse. Potentially because these datasets over-fit, reaching 100% accuracy on training set in all cases.
> Nonetheless, we tried mixture of experts on GCNs (i.e. K=1 but r>1), and it is better than K = r = 1, but not as good as ours using random walks. For example, look at appendix and scroll down, comparing every (K=1,r=4; i.e. mixture of GCN) with (K=4,r=1; ours), and you will find that ours is better in all cases, showing that random walks indeed help.
>
>
> * Calibration is not clean:
>
> You are right. We got excited about the "calibration" paper. Now we removed calibration, as it deviates from our story, which gave us more room to experiment with GraphSAGE (SAGE) and DCNN, and show that we can build a Network of GraphSAGE (N-SAGE).
>
>
> * Does this hold for large datasets?
>
> There are many benchmarks on the datasets we use, including at least a handful of concurrent submissions to ICLR. We said that we will tackle more datasets in "future work". We will do our best to do so by the end of the rebuttal cycle.
>
>
> * Experiments with GraphSAGE and MoNet?
>
> We added experiments with GraphSAGE and DCNN. Our models still outperform. We plan to try-out MoNet but perhaps after adding another (larger) dataset.
>
>
> * Not much novelty?
>
> As we added experiments for GraphSAGE (SAGE), we decided to wrap SAGE in a network and train it with random walks, showing that Network of SAGE (N-SAGE) is better than SAGE. We feel that this generalization makes our work novel enough, and we hope that you agree.

---

> ### Author Response · Authors · 2018-01-06
> **Added PPI dataset, large dataset used by GraphSAGE**
>
> We added experiments for PPI dataset (we downloaded it from the GraphSAGE paper).
>
> The PPI dataset has about ~20 times more edges than our previously-largest dataset.
>
> from SAGE authors:
> SAGE-LSTM gets 61.2
> SAGE [i.e. pooling] gets 60.0
>
> Our implementation of SAGE gets 59.8
> Our method (N-SAGE) gets 65.0
>
> This shows (unmodified) random walk indeed help increase performance. Results are added to the table.

---

### Public Comment · ~Thomas_N._Kipf1 · 2017-11-20
**Validation set and comparison to GraphSAGE**

Very interesting work!

I very much appreciate that you pointed out a significant issue with the benchmark dataset splits for Cora/Citeseer/Pubmed that are now often being used to compare models for semi-supervised learning on graph-structured data. Following https://arxiv.org/abs/1609.02907, the setting is typically as follows (as you mentioned): a small number of labeled examples are used for training (typically 20 labeled nodes per class), whereas a large fixed-size split of 500 labeled nodes is used for validation / hyperparameter optimization.

While the hyperparameter optimization procedure in https://arxiv.org/abs/1609.02907 was kept to a bare minimum (same hyperparameter choice across all three datasets, chosen from a very small grid search on Cora), it is indeed possible to easily "cheat" the benchmark by making use of the rich information provided by the validation set, as your results denoted by 'calibrated' (where you perform gradient descent on some of the model parameters based on validation loss) nicely demonstrate. I am a bit worried that this issue affects a number of recently proposed models that make use of this benchmark (some of the other concurrent submissions to ICLR2018 included), as it is hard to evaluate how much effort has been put into hyperparameter optimization.

It looks to me like your uncalibrated model (i.e. without gradient descent optimization based on validation loss) is unaffected by this and indicates that your proposed Network of GCNs indeed helps improve model performance.

Recently, a number of improvements of the GCN model have been proposed, and I think it would make your results stronger if you compared to the most prominent ones that have been published lately: GraphSAGE (https://arxiv.org/abs/1706.02216) and mixture model CNNs (https://arxiv.org/abs/1611.08402). Arguably their contributions are orthogonal to yours, so ideally these model improvements could easily be combined. Nonetheless it would provide a clearer picture to see how different contributions can make this class of model more powerful / stable.

It would be interesting to see an evaluation of your model on at least one different type of dataset (such as one of the benchmark datasets introduced in GraphSAGE), where calibration on the validation set hopefully wouldn't have such a large impact.

---

> ### Author Response · Authors · 2017-11-22
> **RE: Validation set and comparison to GraphSAGE**
>
> Thomas,
>
> Thanks for your kind words about our work!
>
> We share your feelings about the challenges of assessing work that uses the benchmark splits, and we agree that testing on more datasets (e.g. graphs introduced in GraphSAGE) would further test if our model can generalize to other settings which hopefully do not suffer from the train VS validation size variance.
>
> We were not aware of GraphSAGE at the time of our work (it is recent, to appear in NIPS). Nonetheless, it should be a one-line addition to our baseline (Kipf's GCN) and our model (NGCN), as it is just a layer-norm transformation (https://arxiv.org/abs/1607.06450).
>
> We hope to add some additional experimental results during the rebuttal phase, as we are also quite interested in understanding the impact of newer models (e.g. GraphSAGE and/or mixture of CNNs) in the context of our proposed method. This should strengthen our work -- Thank you for the suggestion!

---

> ### Author Response · Authors · 2017-12-24
> **Added GraphSAGE and a Network of GraphSAGE (N-SAGE)**
>
> Thomas,
>
> We now added experiments to GraphSAGE and also to Network of GraphSAGE. We only used one version of GraphSAGE, which is the mean pooling aggregation, as the authors of GraphSAGE mention that it performs on-par with their max-pooling aggregation model -- we did not try their LSTM aggregation.
>
>
> TLDR:
>
> * GraphSAGE performs better than GCN, when training data is very scarce (e.g. 5 or 10 labeled nodes per class).
>
> * GCN out-performs GraphSAGE with more training data (e.g. >= 20 labeled nodes per class).
>
> * Network of GraphSAGE (N-SAGE) is better than GraphSAGE in all scenarios.

---

> > ### Public Comment · ~Thomas_N._Kipf1 · 2018-01-02
> > **GraphSAGE and a Network of GraphSAGE (N-SAGE)**
> >
> > Thanks for adding this explicit comparison! It looks like N-SAGE can improve upon the GraphSAGE results by quite a bit.

---

### Decision · Program_Chairs · 2018-01-29
**ICLR 2018 Conference Acceptance Decision**

**Decision:**

Reject

**Comment:**

This paper proposes a multiscale variant of Graph Convolutional Networks (GCN) , obtained by combining separate GCN modules using powers of normalized adjacency as generators. The model is tested on several node classification semi-supervised tasks obtaining excellent numerical performance.

Reviewers acknowledged the good empirical performance of the model, but all raised the issue of limited novelty, relative to the growing body of literature on graph neural networks. In particular, they missed an analysis that compares random walks powers to other multiscale approaches and justifies its performance in the context of semi-supervised learning. Overall, the AC believes this is a good paper, but it can be significantly stronger with an extra iteration that addresses these limitations.